# Assessing the Quality and Coverage of Maternal Postnatal Care in Bangladesh: A Comparative Analysis of Quality Postnatal Care among Home and Facility Births

**DOI:** 10.3390/ijerph21030359

**Published:** 2024-03-18

**Authors:** Sabrina Sharmin Priyanka, Dibbya Pravas Dasgupta, Abu Yousuf Md Abdullah, Nazia Binte Ali, Hafeza Khatun, Sk Masum Billah

**Affiliations:** 1Maternal and Child Health Division, International Centre for Diarrhoeal Disease Research, Bangladesh (icddr,b), Dhaka 1000, Bangladesh; naziabinteali@g.harvard.edu (N.B.A.); billah@icddrb.org (S.M.B.); 2Centre for Health Services and Policy Research, School of Population and Public Health, The University of British Columbia, Vancouver, BC V6T 1Z4, Canada; dibbya@student.ubc.ca; 3School of Planning, University of Waterloo, Waterloo, ON N2L 3G1, Canada; aymabdullah@uwaterloo.ca; 4Global Health and Population, Harvard T. H. Chan School of Public Health, Harvard University, 677 Huntington Ave, Boston, MA 02115, USA; 5Binary Data Lab, Dhaka 1208, Bangladesh; hafeza@binarydatalab.org; 6Sydney School of Public Health, The University of Sydney, Sydney, NSW 2006, Australia

**Keywords:** quality postnatal care, home births, facility births, postnatal care components, maternal postpartum deaths, Bangladesh maternal mortality survey

## Abstract

Background: Bangladesh has achieved remarkable progress in reducing maternal mortality, yet postpartum deaths remain a significant issue. Emphasis on quality postnatal care (qPNC) is crucial, as increased coverage alone has not sufficiently reduced maternal morbidity and mortality. Methods: This study included data from the Bangladesh Maternal Mortality Survey of 32,106 mothers who delivered within three years prior to the survey. Descriptive statistics were used to report coverage and components of postnatal care stratified by covariates. Log-linear regression models were used to assess the determinants of quality postnatal care among facility and home births. Results: From 2010 to 2016, postnatal care coverage within 48 h of delivery by a qualified provider rose from 23% to 47%. Of the births, 94% were facility births that received timely PNC, contrasted with only 6% for home births. Despite the increased coverage, quality of care remained as low as 1% for home births and 13% for facility births. Key factors affecting qPNC utilization included socio-demographic factors, pregnancy complications, type of birth attendant, delivery method, and financial readiness. Conclusion: Importantly, deliveries assisted by skilled birth attendants correlated with higher quality postnatal care. This study reveals a significant gap between the coverage and quality of postnatal care in rural Bangladesh, especially for home births. It underscores the need for targeted interventions to enhance qPNC.

## 1. Introduction

Globally, an estimated 810 women die every day from preventable causes related to pregnancy and childbirth; of these deaths, 94% of maternal deaths occur in low-resource settings [1]. Half of all maternal and newborn deaths occur within the first 24 h after birth, and, among these, 75% of newborns die within the seven days following delivery [2]. Most of these deaths occurring immediately after birth can be averted by providing appropriate and timely postnatal care [3,4,5]. Studies in South Africa and Asia estimated that maternal deaths are extremely high within the intrapartum period and up to the first two days of childbirth [6,7]. Additionally, studies in 75 high-burden countries suggest that, by 2025, focusing on quality of care and enhancing interventions from preconception through postnatal stages could substantially reduce 54% of maternal deaths and 71% of neonatal deaths annually [8,9]. The World Health Organization’s (WHO) postnatal care guideline recommends that women must receive at least one postnatal checkup within the first 48 h of childbirth, along with the recommended postnatal components [10,11].

Since 2000, Bangladesh has made remarkable progress in reducing maternal mortality. However, this progress has stagnated, as the maternal mortality ratio of 2010 and 2016 are almost identical (196 per 100,000 live births) [12,13]. This stall in the decline of maternal mortality is alarming, as nearly 70% of these deaths occur during the postpartum period, accounting for nearly 135 deaths per 100,000 live births, and this trend remained steady across the BMMS surveys conducted in 2001, 2010, and 2016. [14]. The leading cause of all these maternal deaths are ante and postpartum hemorrhage (31%), followed by eclampsia (24%) [14]. In Bangladesh, nearly 88% of women were found to have received antenatal care (ANC) from a skilled provider at least once during pregnancy, but only 21% received quality care [15]. In comparison, the proportion of women who were delivered by a skilled provider and had postnatal care within two days after birth was lower, at 50% and 52%, respectively [15]. National surveys included the quality of antenatal care; however, no data for Bangladesh is available that highlights the quality of postnatal care. Quality care in the context of postnatal care (PNC) signifies the provision of necessary medical attention and support to both mothers and newborns after childbirth, in compliance with established healthcare guidelines and practices [11]. Quality care aims to improve health outcomes by preventing complications, detecting and treating health issues immediately, and providing a positive experience for the mother and child [10]. The fourth Health Nutrition and Population Sector Program (HNPSP) of Bangladesh recommended initiating PNC within 48 h of birth, which is also considered as early PNC [16]. However, this focuses only on connecting mothers and newborns with the health system, with no suggestion on the quality of services received during that time. In spite of undertaking several quality of care initiatives by the Bangladesh Government, substandard quality continues to be a cause of a growing concern [17].

Evidence indicated that increased recommended postnatal coverage—which is also the proportion of women who utilized postnatal care services—alone does not reduce maternal morbidity and mortality [8]. The introduction and implementation of quality care are critical in maternal and newborn care [10,11]. Baqui et al. (2009) showed that receiving the first postnatal visit on the day of birth in Bangladesh was associated with considerably lower neonatal mortality than receiving no visit after birth [18]. Although several studies used secondary analysis to highlight factors influencing women’s uptake of postnatal care, most of these studies ignored the components and quality of postnatal care. With the expansion of coverage, the ongoing concern over poor quality becomes more evident, particularly in terms of inadequate patient–provider interaction, lack of necessary healthcare services, and insufficient use of medical guidelines, which can lead to adverse health outcomes [19,20]. Increasing coverage without quality leads to missed opportunities to improve outcomes, which results in unsafe or delayed procedures and treatments [17]. Thus, there is a dire need to further explore the gap between coverage and quality of services received by women that may influence postnatal care utilization within 48 h of birth [21,22,23,24].

For this study, we have utilized data from national-level household surveys, the Bangladesh Maternal Mortality Surveys (BMMS), to study the changes in the prevalence of coverage and content of postnatal care and explored the determinants of quality postnatal care. To measure the quality, we have followed the WHO’s standards of improving the quality of maternal and newborn care [11]. However, even surveys as extensive as BMMS do not collect all components of quality postnatal care as recommended by the WHO. As there is a scarcity of evidence addressing determinants of quality postnatal care (qPNC) practices of mothers in the current country context, the objective of the proposed study is to examine and establish the gap between coverage and components as a proxy for quality postnatal care and to assess the range of factors that affects the quality postnatal care utilization following childbirth by medically trained providers among facility and home births in Bangladesh.

## 2. Materials and Methods

### 2.1. Study Data

This study focused on the survey area in Bangladesh covered by the BMMS 2010 and 2016 [12,13]. BMMS 2010 interviewed around 175,261 women from 168,629 households. BMMS 2016 is one of the largest national level surveys in Bangladesh, engaging over 113,633 women from around 300,000 households to discuss birth outcomes over the past three years, surpassing the scope of using any other dataset and offering nationwide insights. The primary objective of these surveys was to collect data on maternal mortalities in Bangladesh. The required number of households to be surveyed was calculated using a multi-stage sample selection procedure, the details of which could be found elsewhere [14]. Three types of regions were covered including urban, rural, and other (peri-) urban areas. At the household level (household questionnaire), the questionnaires focused on household background characteristics and death information, and at the individual level (women’s questionnaire), they focused on respondents’ background, reproductively, child mortality, and family planning.

Additional questionnaires, such as the verbal autopsy and community questionnaires, collected information on causes of death for female adults preceding the survey, and the latter queried about the socioeconomic condition of the community and the availability and accessibility of health and family planning in the community [14].

### 2.2. Study Design and Settings

Since the primary outcome of interest of our study was PNC and its components for the most recent live births, we limited the sample to women who were asked if they received postnatal checkups for themselves or their babies during the two months following delivery for the most recent live births in the three years before the BMMS 2016 survey. For comparisons and descriptive analysis, we have used BMMS 2010, and for determinants analysis, we have used BMMS 2016. We have applied similar selection criteria for the BMMS 2010 survey data as of BMMS 2016. The total sample size for BMMS 2010 was 17,149 and for BMMS 2016 was 32,106 mothers (Figure 1).

### 2.3. Outcome Variable

To define qPNC, we used the WHO recommendations on postnatal care of the mother and newborn [10,25]. A postnatal care visit was considered a ‘qPNC’ if it had covered the any of the following at least once during the postnatal period:At least one PNC within 48 h by a medically trained provider (MTP), who could be a doctor, nurse/midwife/paramedic/family welfare visitor (FWV), community skill birth attendant (CSBA), and sub assistant community medical officer (SACMO);Breast examination;Counseled on postpartum danger signs;Temperature check;Checked for vaginal discharge (to monitor excessive bleeding and foul-smelling discharge).

### 2.4. Key Independent Variables

Key independent variables described below were chosen based on previous literature reviews.

#### 2.4.1. Background Characteristics

The mother’s age at birth (<20, 20–34, and 35–49, education (no education, primary, and secondary or above), religion (Muslim and others), household wealth (poor, middle class, and rich), birth order (1, 2, and 3 or more), and ownership of mobile phone (yes or no) were analyzed as categorical variables. Due to their low numbers, all religious statuses other than Muslim were combined into one category and labeled “others”.

#### 2.4.2. Maternal Health Services

In terms of maternal health services, the following indicators were considered:ANC from MTP: no ANC, ANC from qualified (which includes doctors, Nurses/midwives/paramedics/FWV, CSBA, MA/SACMO), and unqualified providersNumber of ANC: no ANC, 1–3 ANC, and 4 or more ANC;Place of delivery: home and facility birthsType of birth attendant: skilled and unskilled (skilled providers include qualified doctors, nurses/midwives/paramedics/FWV, CSBA, and MA/SACMO);Mode of delivery: normal or c-section;Complications during pregnancy: yes or no;Complications during delivery: yes or no;Complications during the postnatal period: yes or no;Savings available for delivery care: yes or no;Mobile phone ownership: yes or no.

### 2.5. Statistical Analysis

Weighted descriptive statistics and chi-square analyses were carried out on all predictor and demographic variables from the BMMS 2016 survey to compare PNC provided by qualified healthcare providers within 48 h of home and health facilities delivery. Additionally, from both BMMS 2010 and BMMS 2016, descriptive statistics were calculated for the components of PNC. For creating the categories of household wealth, we used the data on ownership of household items. We used the data on household possessions; floor construction materials, wall, and roof; drinking water source; toilet facilities; and ownership of land and domestic animals. Through principal components analysis, each asset was assigned a weight (factor score). After summing each household score, individuals were ranked into three wealth categories (poor, middle class, and rich), according to the total score of the household where they live.

Log-linear regression models predicting the receipt of qPNC outcomes among home and facility deliveries were developed, using predicting factors such as mother’s education, wealth index, number of ANC, ANC from MTP, type of birth attendant, complication during pregnancy, complication during delivery, complication during postnatal period, and savings for delivery care. In log-linear regression models predicting the receipt of qPNC outcomes among facility births, we added type of facility delivery and the mode of delivery with the mentioned variables, as these variables were not relevant for home births. Stata 14.2 was used to analyze the data. Since we utilized a multi-stage cluster sampling methodology, all estimates were produced following the requisite weighting. Statistically significant associations were determined based on *p*-value < 0.05.

### 2.6. Ethics

The study utilized publicly accessible secondary data provided by measure evaluation and the Bangladesh Maternal Mortality Survey (BMMS). No ethical approval was necessary for the use of this secondary data, as they fall under the category of publicly available datasets, and the data usage complies with the guidelines provided by the data source. Comprehensive details regarding the data collection methods used by BMMS are documented in published reports [12,13]. 

## 3. Results

### 3.1. Background Characteristics of Study Population

This study included a total weighted sample of 32,106 reproductive aged mothers from all eight division of Bangladesh. Of the women in our sample, the majority (78%) were aged between 20 and 34, and 91% had access to a mobile phone, 61% completed secondary education or higher, and 42% belonged to a poor household. Only 38% of the women received four or more ANC, and half (50%) of the mothers were delivered by skill birth attendants (Table 1).

### 3.2. Components of PNC

There was a noticeable difference in the type of PNC received by women at home compared to those in healthcare facilities. Figure 2 highlights that, for births in facilities, the majority (94%) received PNC within two days from medical professionals, and 85.3% of women had their temperature checked. In contrast, for home births, only 5.8% received PNC within two days from a health professional, and about one third had their temperature measured. The least common practice was checking for vaginal discharge, with only 10.9% of home births and 32.6% of facility births undergoing this check. The overall prevalence of qPNC was very low at only 6.6%.

In 2016, 44.6% of women who had home deliveries and 95.9% of women having facility deliveries had received a postnatal visit within 42 days of the delivery. In contrast, only 0.6% and 13.3% of women had received qPNC for home and facility births, respectively (Table 2). Between 2010 and 2016, there was an increase in receiving a postnatal visit within 48 h and receiving this visit from a qualified provider. Additionally, the highest percentage of increase from 2010 to 2016 was observed for receiving a PNC within 48 h (34%), followed by receiving a PNC within 42 days (28%) and receiving a PNC visit by a qualified provider within 48 h (24%).

### 3.3. Utilization and Factors Associated with qPNC in Bangladesh

Table 3 reports the prevalence and factors associated with qPNC among home and facility births; with facility births showing higher prevalence rates compared to home births across various factors. For example, younger mothers and those with no education had lower rates of qPNC, while wealthier mothers and those who delivered in facilities showed higher rates. The table suggests that having skilled birth attendants was a strong precursor of qPNC for both facility (13.6%) and home deliveries (6.3%). Similarly, the tables also demonstrate that the percentages of women receiving qPNC after facility births and reporting no complications during delivery (15.3%) and postnatal period (15.5%) were higher compared to women who received qPNC after home births (1.2% and 1.4%, respectively).

### 3.4. Determinants of qPNC in Bangladesh

Logistic regression analysis for mothers who had home births indicates that, compared to a poor household, a mother from a rich household is 1.95 times (aRR: 1.95, 95% CI [1.22, 3.11]) more likely to get quality postnatal care. Compared to mothers with unskilled birth attendants, mothers with skilled birth attendants are 19.8 times more likely (aRR: 19.8, 95% CI [13.04–30.05]) to receive qPNC. Mothers with a complication during a postnatal period were 2.48 times more likely (aRR: 2.48, 95% CI [1.60, 3.84]) to get quality postnatal care. In contrast, no significant association was found for mothers’ age at birth, education, number of ANC, ANC from MTP, saving money for delivery care, and having a mobile phone.

Furthermore, Table 4 also illustrates determinants of the qPNC of mothers who had facility births. Mothers whose age at birth was 20 to 34 years had 17% more odds (aRR: 1.17, 95%CI [1.04–1.32]) of receiving qPNC than mothers whose age at birth was less than 20 years old. Compared to mothers with unskilled birth attendants, mothers with skilled birth attendants are 7.90 times more likely (aRR: 7.90, 95% CI [1.98–31.41]) to receive qPNC. Additionally, compared to normal delivery, mothers who had c-sections are 1.58 times more likely (aRR: 1.58, 95% CI [1.42–1.76]) to get quality postnatal care. Moreover, complications during delivery and savings available for delivery care are significantly associated (aRR: 1.21, 95% CI [1.10–1.33] and aRR: 1.20, 95% CI [1.10–1.31]) with receiving qPNC. No significant associations were found for the mother’s age at birth, mother’s education, wealth index, number of ANC, ANC from MTP, type of facility births, complications during the postnatal period, and owning a mobile phone.

## 4. Discussion

This study utilized national, cross-sectional surveys to identify the factors influencing the quality of postnatal care (qPNC) for both home and facility births within the context of Bangladesh, emphasizing the essential components of PNC and their impact on the likelihood of women receiving high quality care. A notable finding was the gap between the coverage and quality of postnatal care, with a significant majority of women not receiving quality care. This discrepancy aligns with findings from a similar study in Myanmar, which underscores that, although women birthing in a facility had adequate contacts through qualified providers, women who gave birth at home had a very low prevalence of postnatal contact by qualified providers within 48 h after birth [26]. Results from our study should supplement the existing evidence by identifying potential gaps and opportunities for improving quality postnatal care practices in rural Bangladesh.

The analysis identified sociodemographic factors and postnatal complications as significant determinants of qPNC for home births, while, for facility births, factors such as age, delivery method, delivery complications, and financial preparedness for delivery were significantly associated with quality care. Despite an increasing trend in postnatal check-ups from 23% in 2010 to 47% in 2016 by qualified providers within 48 h after birth, the prevalence of qPNC remained alarmingly low at 6.6% in 2016. This stark gap between contact and quality underlines the critical need for improvement in postnatal care services.

Several studies have indicated that increased recommended postnatal coverage alone does not reduce maternal morbidity and mortality [8]. The introduction and implementation of quality of care is recognized as a critical aspect in maternal and newborn care [10,11,25]. Our analysis underlined that, in spite of this, the quality of postnatal care remains as low as 1% for home birth and 13% for facility births. This study further revealed that the distribution of PNC components demonstrated a wide variance in the quality spectrum of postnatal care, with significant deficiencies in basic health checks and counseling services. For instance, slightly more than half of the women had their temperature checked (57%). Similarly, more focus should be given to providing counselling on postpartum maternal and newborn danger signs (27%) and examining vaginal discharge (21%) after delivery to detect post-partum hemorrhage or infection. Mothers’ knowledge of danger signs can help identify early maternal and newborn complications, highlighting the necessity for enhanced educational efforts and service delivery. Components of postnatal care for women who had home delivery were low across all spectrums.

For home births, both the coverage and quality of postnatal care were found to be strikingly low. Previous studies highlighted the importance of early initiation of PNC, which is receiving PNC within 48 h of birth [18]. This period is critical for monitoring the health of both the mother and the newborn, as it helps to identify and address any immediate health concerns and ensures the well-being of both parties during the initial post-birth phase. Our findings showed only 5.8% of women received postnatal care from a medically trained provider within two days post-delivery. There might be cultural and religious attributes to lower postnatal care. The sociocultural beliefs in the country can hinder women’s access to healthcare services, particularly those provided by male or unfamiliar providers [23]. Similar studies in the country highlighted that women from Muslim households need permission from their husbands to go to health facilities [27]. All these barriers point towards implementing policies to deliver postnatal care for home births to avert maternal and neonatal morbidity. Despite the lack of a direct correlation between religion and qPNC, socioeconomic status emerged as a critical factor, with women from higher social classes being significantly more likely to receive quality care.

Wealth was considered to be an important factor for receiving quality care for a home birth, as the cost of care acts as a barrier and hinders access to health care [28]. Our study portrays that mothers who belong to higher social classes were 1.95 times more likely to receive quality postnatal care compared to poor households. Previous studies also showed that mothers who belong to a higher social class were more likely to seek qPNC during maternal complications [23,29]. Our findings underscore the importance of antenatal care as a pivotal point for educating mothers on maternal danger signs and postpartum complications, promoting a continuum of care [30,31,32]. Studies in India and Cambodia found that women receiving high-quality antenatal care were better informed about pregnancy during ANC visits, were more likely to be delivered by skilled birth attendants, and continued receiving postnatal care after childbirth [33,34]. Our findings failed to find such an association.

Women with complications during the postnatal period were 2.48 times more likely to receive quality postnatal care. Complications during the postnatal period were a significant predictor for seeking qPNC, indicating that routine postnatal care is often overlooked unless complications arise. A study in Pakistan also demonstrated that mothers visit health facilities during postnatal periods for serious and fatal complications only [35]. This reactive approach to postnatal care highlights the need for proactive and routine postnatal services to identify and manage postpartum complications promptly. Despite having several home visitation programs provided by the Government of Bangladesh, postnatal care remains low. Therefore, policies should concentrate more on ensuring home visits at the community level and improving high-qPNC packages for reaching women.

The place of delivery has always played a vital role in obtaining maternal postnatal services [36]. In our study, 94% of women who had facility births received PNC within 48 h of birth. Among them, older women had 17% higher use of qPNC. Age is often used as a proxy for mothers’ accumulated knowledge of healthcare services, which may guide them to access healthcare services [21].

Factors such as cesarean delivery and delivery complications were positively associated with the likelihood of receiving qPNC, reflecting the heightened risk profile of these groups [37,38]. Despite higher coverage of PNC for women having facility births, such low qPNC at facilities can be explained by mothers’ short hospital stays, especially following vaginal deliveries, and this may contribute to the low incidence of qPNC at facilities [39]. However, there is a need for future studies to include data on the duration of hospital stays to better understand PNC dynamics. Our findings suggested that women who reported saving money for delivery were more likely to obtain quality care for facility births.

Women who were attended by skilled birth attendants during delivery exhibited a higher probability of receiving quality postnatal care, applicable to both home and facility births. This suggests that skilled attendants play a crucial role in providing necessary health information, improving post-birth care by promoting health assessments and identifying early complications, thus enhancing the overall quality of postnatal care received by mothers. Studies in 30 lower-middle-income countries had also reported such associations [39,40]. This correlation suggests that the presence of skilled personnel during childbirth sets a foundational standard for subsequent quality care, highlighting the need for comprehensive training and the deployment of skilled birth attendants across all birthing environments [41,42].

There were some limitations of this study. A cross-sectional survey was used to gather the data for this study’s outcomes; as a result, we were unable to draw a connection between the explanatory factors and relevant PNC-related outcomes of interest. Additionally, the survey collected retrospective data based on the information provided by the respondent, which may be subjected to recall bias. Moreover, despite our preference for more recent datasets, the BMMS 2016 was the most recent available dataset that captured the targeted variables. Other important components that can be used to measure quality postnatal care, like blood pressure measurement, measurement of anemia, checking urine, and asking for other vital signs, were not collected in this survey.

## 5. Conclusions

Ensuring recommended postnatal care has the potential to save maternal and newborn lives after birth. Our study indicated an alarming difference between coverage and components of the quality of postnatal care. Therefore, this is the time when our policies need to focus on components of postnatal care. National surveys should include the spectrum of quality postnatal care according to the WHO’s qPNC guidelines to better understand the complete scenario of quality postnatal care. Need-based solutions need to be considered as well when planning for interventions, which should also determine health resource allocations in the country. Furthermore, there is a critical need to amplify awareness and counseling initiatives regarding the benefits of postnatal care to encourage facility returns. Given the high rate of home births, the integration of quality postnatal care (qPNC) into home visit programs is essential. As the Government of Bangladesh continues to invest in community-level health initiatives, a significant focus should be placed on educating about the significance of postpartum care to ensure comprehensive and effective service delivery. The existing system should ensure all the services are readily available and rightly provided to mothers and their newborns.

## Figures and Tables

**Figure 1 ijerph-21-00359-f001:**
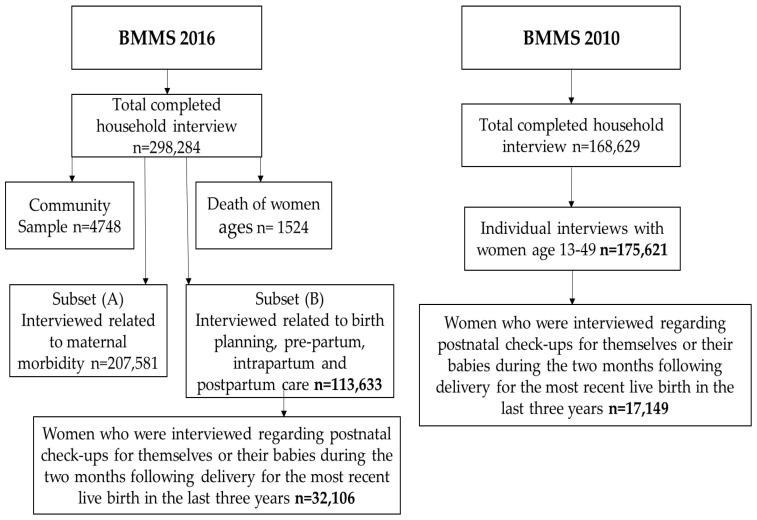
Flow chart of study population included in the analysis.

**Figure 2 ijerph-21-00359-f002:**
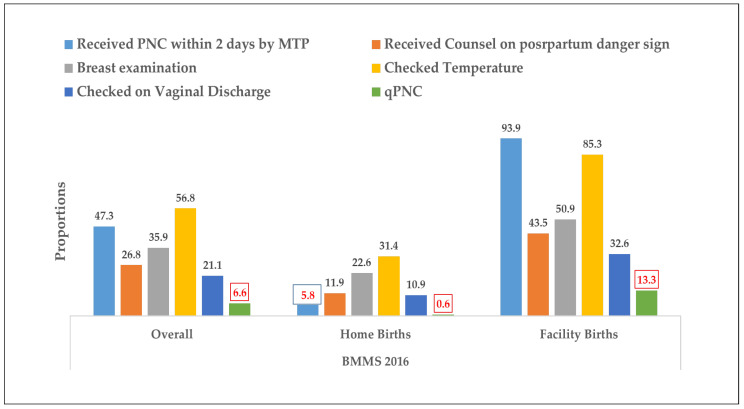
Prevalence of PNC components during BMMS 2016.

**Table 1 ijerph-21-00359-t001:** Sample characteristics of BMMS 2016 (n = 32,106).

Characteristic	n	%
**Mother’s age at birth**		
<20	5094	15.9
20–34	24,894	77.5
35–49	2118	6.6
**Mobile phone**		
Yes	29,239	91.1
No	2867	8.9
**Mother’s education**		
No education	3145	9.8
Primary	9487	29.6
Secondary+	19,474	60.7
**Religion**		
Muslim	29,517	91.9
Others	2589	8.1
**Wealth index**		
Poor	13,344	41.6
Middle class	6287	19.6
Rich	12,475	38.9
**Birth Order**		
1	13,358	41.6
2	11,067	34.5
3+	7681	23.9
**Number of ANC**		
No ANC	5602	17.5
1–3	14,292	44.5
4+	12,212	38.0
**ANC from MTP**		
No ANC	5608	17.5
Qualified Provider	23,581	73.5
Unqualified Provider	2917	9.1
**Type of facility of delivery**		
Private facility	10,341	67.7
Public facility	4936	32.3
**Type of birth attendant**		
Skilled	16,176	50.4
Unskilled	15,930	49.6
**Complications during** **pregnancy**		
Yes	20,379	63.5
No	11,727	36.5
**Complications during** **delivery**		
Yes	24,179	75.3
No	7927	24.7
**Complications during** **Postnatal Period**		
Yes	25,684	80.0
No	6422	20.0
**Savings available for delivery care**		
Yes	14,894	46.4
No	17,212	53.6

**Table 2 ijerph-21-00359-t002:** Prevalence of postnatal care-related indicators among home and facility births.

	2010			2016		
Indicators	Home Births	Facility Births	Total	Home Births	Facility Births	Total
	%					
Received postnatal visit within 42 days	-	-	41.0	44.6	95.9	69.0
Received postnatal visit within 48 h	16.6	80.1	32.0	39.7	94.9	66.0
Postnatal visit by a qualified provider within 2 days	4.6	78.8	23.0	5.8	93.9	47.0
qPNC	-	-		0.6	13.3	6.6

BMMS survey data was used for 2010 and 2016.

**Table 3 ijerph-21-00359-t003:** Prevalence and factors associated with qPNC among home and facility births during BMMS 2016.

Covariates	qPNC among Facility Births (n = 15,277)			qPNC among Home Births (n = 16,829)		
	n	%	*p*-Value	n	%	*p*-Value
Mother’s age at birth						
<20	270	11.2	0.002	13	0.5	0.259
20–34	1652	13.8		81	0.6	
35–49	133	14.1		11	0.9	
Mobile phone			0.847			0.320
Yes	1883	13.4		92	0.6	
No	172	13.6		13	0.8	
Mother’s education						
No education	87	11.2	<0.001	9	0.4	0.017
Primary	341	11.3		32	0.5	
secondary+	1627	14.2		64	0.8	
Religion			0.809			0.252
Muslim	1838	13.4		96	0.6	
Others	217	13.6		9	0.9	
Wealth index						
Poor	480	12.3	<0.001	32	0.3	<0.001
Middle class	351	11.8		21	0.6	
Rich	1224	14.6		52	1.3	
Birth Order						
1	1041	13.6	0.345	40	0.7	0.640
2	709	13.7		35	0.6	
3+	305	12.5		30	0.6	
Number of ANC						
No ANC	77	10.6	<0.001	12	0.2	<0.001
1–3	762	11.9		50	0.6	
4+	1216	15		43	1.1	
ANC from MTP						
No ANC	75	10.3	0.001	12	0.2	<0.001
Qualified Provider	1913	13.8		86	0.9	
Unqualified Provider	67	10		7	0.3	
Type of facility of delivery						
Private facility	1495	14.5	<0.001			
Public facility	560	11.3				
Type of birth attendant						
Skilled	2053	13.6	<0.001	67	6.3	<0.001
Unskilled	2	1.2		38	0.2	
Complications during pregnancy						
Yes	1104	12.7	0.003	62	0.5	0.019
No	951	14.4		43	0.8	
Complications during delivery						
Yes	1357	12.7	<0.001	65	0.5	<0.001
No	698	15.3		40	1.2	
Complications during Postnatal Period						
Yes	1520	12.8	<0.001	63	0.5	<0.001
No	535	15.5		42	1.4	
Savings available for delivery care						
Yes	635	11.4	<0.001	43	0.5	0.003
No	1420	14.6		62	0.8	

**Table 4 ijerph-21-00359-t004:** Estimated unadjusted (uRR) and adjusted risk ratio (aRR) for multivariate logistic regression models of qPNC among home (n = 16,829) and facility birth (n = 15,277).

Covariates	qPNC among Home Births				qPNC among Facility Births			
	uRR (95% CI)		aRR (95% CI)		uRR (95% CI)		aRR (95% CI)	
Mother’s age at birth								
<20	1.00		1.00		1.00		1.00	
20–34	1.29 (0.72–2.32)		1.22 (0.69–2.18)		1.23 (1.09–1.39)	**	1.17 (1.04–1.32)	*
35–49	1.93 (0.87–4.31)		2.02 (0.9–4.54)		1.25 (1.04–1.53)	*	1.19 (0.98–1.44)	
Mother’s education								
No education	1.00		1.00		1.00		1.00	
Primary	1.30 (0.62–2.72)		1.19 (0.58–2.45)		1.01 (0.81–1.26)		0.98 (0.79–1.23)	
secondary+	2.11 (1.05–4.24)	*	0.98 (0.48–2)		1.26 (1.03–1.54)	*	1.08 (0.88–1.33)	
Wealth index								
Poor	1.00		1.00		1.00		1.00	
Middle	1.86 (1.08–3.23)	*	1.34 (0.77–2.33)		0.96 (0.85–1.1)		0.88 (0.77–1.00)	
Rich	3.77 (2.43–5.85)	***	1.95 (1.22–3.11)	**	1.18 (1.07–1.31)	**	0.97 (0.87–1.08)	
Number of ANC								
No ANC	1.00		1.00		1.00		1.00	
1–3	2.58 (1.38–4.84)	**	0.59 (0–1.33)		1.12 (0.9–1.4)		0.4 (0.15–1.04)	
4+	4.27 (2.25–8.08)	***	0.74 (0–1.68)		1.42 (1.14–1.76)	**	0.47 (0.18–1.22)	
ANC from MTP								
No ANC	1.00		1.00		1.00		1.00	
Qualified Provider	3.6 (1.97–6.58)	***	2.32 (0–5.23)		1.34 (1.08–1.67)	**	2.51 (0.95–6.65)	
Unqualified Provider	1.27 (0.50–3.22)		1.77 (0–4.03)		0.97 (0.71–1.32)		2.22 (0.82–6.01)	
Type of facility of delivery								
Private facility	-		-		1.00		1.00	
Public facility	-		-		0.78 (0.72–0.86)	***	0.98 (0.89–1.08)	
Type of birth attendant								
Unskilled	1.00		1.00		1.00		1.00	
Skilled	25.92 (17.49–38.40)	***	19.8 (13.04–30.05)	***	11.76 (2.96–46.66)	***	7.9 (1.98–31.41)	**
Mode of delivery								
Normal	-		-		1.00		1.00	
C-section	-		-		1.72 (1.56–1.89)	***	1.58 (1.42–1.76)	***
Complications during delivery								
No	1.00		1.00		1.00		1.00	
Yes	2.46 (1.66–3.63)	***	1.39 (0.9–2.16)		1.21 (1.11–1.32)	***	1.21 (1.1–1.33)	***
Complications during postnatal period								
No	1.00		1.00		1.00		1.00	
Yes	3.11 (2.11–4.58)	***	2.48 (1.6–3.84)	***	1.21 (1.1–1.32)	***	1.08 (0.98–1.19)	
Savings available for delivery care								
No	1.00		1.00		1.00		1.00	
Yes	1.8 (1.22–2.66)	**	1.29 (0.87–1.9)		1.27 (1.17–1.39)	***	1.20 (1.1–1.31)	***
Mobile Phone								
No	1.00		1.00		1.00		1.00	1.00
Yes	0.75 (0.42–1.33)		0.83 (0.47–1.47)		0.99 (0.85–1.14)	***	1.00 (0.87–1.16)	

Note: * *p* < 0.05; ** *p* < 0.01, *** *p* < 0.001.

## Data Availability

The Bangladesh Maternal Mortality and Health Care Survey (BMMS) 2010 and 2016 datasets are publicly available at UNC Dataverse (www.dataverse.unc.edu, accessed on 10 September 2022).

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
