# Peer review of "Assessing the Quality and Coverage of Maternal Postnatal Care in Bangladesh: A Comparative Analysis of Quality Postnatal Care among Home and Facility Births"

_ijerph, 2024, doi:10.3390/ijerph21030359_

Round 1

Reviewer 1 Report

Comments and Suggestions for Authors

Thank you for inviting me to peer review the manuscript “The gap between coverage and components of postnatal care and factors associated with quality postnatal care among home and facility births in Bangladesh.”

The paper under consideration investigates an intriguing topic and tries to advance our understanding of the qPNC among home and facility births in Bangladesh. The study question is well-stated, and the paper's general structure is respectable. However, my primary concern is with the temporal component of the data employed in the study. In fact, the datasets adopted for the study were from 2010 and 2016, raising concerns about their current relevance. Datasets from several years ago may not accurately reflect current conditions. Given the quick improvements and dynamic nature of the area, an update in data collecting would have increased the paper's applicability and reliability.

This is why, in the lack of recent surveys and statistics that can provide a more realistic trend in the care of pregnant women and their newborns, I believe the article should not be accepted for publication.

Author Response

Thank you very much for taking the time to review this manuscript. Please find the detailed responses in the attached document and the corresponding revisions/corrections highlighted/in track changes in the re-submitted files. 

Reviewer 2 Report

Comments and Suggestions for Authors

The article addresses an important and highly relevant issue related to various factors concerning postnatal care in Bangladesh. The conclusions drawn from the article could be of significant assistance for educational purposes as well as for policy regulations. To enhance the clarity and precision of the text, I suggest introducing a few minor changes:

Lines 44-46: The article emphasizes that "studies around the world suggest improving coverage and quality during pre-partum and postpartum periods can reduce 54% of maternal deaths annually by 2025." At this point, the authors refer to only one publication, which pertains to selected countries in Africa. Therefore, it is necessary to either specify which part of the world is being referred to or add relevant literature.

Line 82: The World Health Organization has already been cited earlier. Therefore, it is sufficient to use the abbreviation here. Abbreviations are introduced to facilitate their use in later parts of the text.

Line 210: It appears that the sentence may not have been completed, or the text was modified during file conversion. This needs to be verified.

Lines 349-355: For the clarity of the text, the Discussion should be separated from the Limitations. In short, an additional section titled 'Limitations' should be added.

Line 465: Remove reference number 43 from the bibliography.

Author Response

(The authors gave the same response as above.)
